# A Non-Invasive Method for Monitoring Osteogenesis and Osseointegration Using Near-Infrared Fluorescent Imaging: A Model of Maxilla Implantation in Rats

**DOI:** 10.3390/ijms24055032

**Published:** 2023-03-06

**Authors:** Chien-Chou Lin, Li-Hsuan Chiu, Walter H. Chang, Cheng-An J. Lin, Ruei-Ming Chen, Yuan-Soon Ho, Chun S. Zuo, Austin Changou, Yue-Fa Cheng, Wen-Fu T. Lai

**Affiliations:** 1Graduate Institute of Medical Sciences, College of Medicine, Taipei Medical University, Taipei 110, Taiwan; 2McLean Imaging Center, McLean Hospital and Harvard Medical School, Belmont, MA 02478, USA; 3Department of Biomedical Engineering, Chung Yuan Christian University, Taoyuan 320, Taiwan; 4Ph.D. Program for Translational Medicine, College of Medicine and Technology, Taipei Medical University, Taipei 110, Taiwan; 5Core Facility Center, Office of Research and Development, Taipei Medical University, Taipei 110, Taiwan; 6College of Basic Medicine, North China University of Science and Technology, Tangshan 066008, China; 7Institute of Graduate Clinical Medicine, Taipei Medical University, Taipei 110, Taiwan; 8Department of Research and Department of Dentistry, Taipei Medical University-Shuang-Ho Hospital, New Taipei City 235, Taiwan

**Keywords:** artifact, near-infrared ray, gold nanoparticle, hydroxyapatite, osteoblast

## Abstract

Currently, computed tomography and conventional X-ray radiography usually generate a micro-artifact around metal implants. This metal artifact frequently causes false positive or negative diagnoses of bone maturation or pathological peri-implantitis around implants. In an attempt to repair the artifacts, a highly specific nanoprobe, an osteogenic biomarker, and nano-Au-Pamidronate were designed to monitor the osteogenesis. In total, 12 Sprague Dawley rats were included in the study and could be chategorized in 3 groups: 4 rats in the X-ray and CT group, 4 rats in the NIRF group, and 4 rats in the sham group. A titanium alloy screw was implanted in the anterior hard palate. The X-ray, CT, and NIRF images were taken 28 days after implantation. The X-ray showed that the tissue surrounded the implant tightly; however, a gap of metal artifacts was noted around the interface between dental implants and palatal bone. Compared to the CT image, a fluorescence image was noted around the implant site in the NIRF group. Furthermore, the histological implant-bone tissue also exhibited a significant NIRF signal. In conclusion, this novel NIRF molecular imaging system precisely identifies the image loss caused by metal artifacts and can be applied to monitoring bone maturation around orthopedic implants. In addition, by observing the new bone formation, a new principle and timetable for an implant osseointegrated with bone can be established and a new type of implant fixture or surface treatment can be evaluated using this system.

## 1. Introduction

Radiography imaging of hard tissue is a prevalent diagnostic instrument in dentistry and orthopedics. Intraoral periapical radiography, conventional multi-slice computed tomography, and cone beam computed tomography are lower radiation dose systems commonly used to assess the marginal bone level and detect signs of failing osseointegration around dental implants [1,2]. However, they are susceptible to the appearance of artifacts generated by dental implants [3,4,5,6,7]. In recent years, near-infrared fluorescent (NIRF) imaging technologies have been developed for detecting and monitoring disease progression in joint and bone tissue [8,9,10,11,12]. The techniques also enable clinicians to evaluate the bone tissue healing process around implants at a molecular level. Bone is a metabolically active tissue that can execute remodeling. Remodeling balances osteoblast-induced mineralization and osteoclast-induced demineralization [13]. During this process, hydroxyapatite (HA) is the prime mineral product of osteoblasts and binds to naturally occurring pyrophosphates and phosphonates with a high affinity [14,15,16]. Osteoblasts synthesize HA, which promotes cell adhesion and osteogenic differentiation in human mesenchymal stem cells [17]. Therefore, HA deposition is a marker of bone regeneration and also of the process of cancer formation and atherosclerosis [18,19,20,21].

Monitoring and evaluating osteogenesis and osteointegration around implant objects is proving to be valuable because a variety of dental and orthopedic implants have been utilized in the last 10 years. An ideal monitoring system or method in clinics to determine the optimal force loading time and to achieve the early diagnosis of pathological changes around implants is in high demand [6,7].

Currently, a non-isotopic method does not exist for directly detecting osteoblastic activity in vivo. The wavelength of 700–900 nm fluorescent light, the near-infrared (NIR) “window”, possesses several inherent advantages for the detection of HA compared with ultraviolet and visible light ranges [22,23,24,25]. The NIR photon enables deep penetration into the tissue and subsequentlyexits the tissue easily. Tissue absorption and autofluorescence is minimized, yielding an inherently high contrast between the target and background, and the optical scatter within the tissue is lower [18]. A suitable fluorescence-based optical imaging agent, with emission in the NIR 700–900 nm window, should have in vitro and in vivo stability; resistance to photobleaching; high quantum yield and high absorbency; resistance to metabolic disintegration and nontoxicity; and adequate dispersibility in the biological environment [26]. Among numerous nanoparticles, Au nanoparticles (nano-Aus) not only correspond to the aforementioned criteria but their surface functionalization with molecular groups also exhibits calcium affinity that can enable the targeted delivery of nano-Aus to calcified tissue, including damaged bone tissue [27,28].

To evaluate osteogenesis and osseointegration, we designed a molecular imaging system, Taiwan 1 (TW1)-NIRF, including a new light detection; a charge-coupled device camera; and an HA binding reagent, pamidronate (Pam), which can rapidly conjugate to nano-Au particles to form a probe, nano-Au pamidronate (nano-Au-Pam). With the nano-Au-Pam probe, the camera enables the identification osteoblast-synthesized HA. The osteoblast-synthesized HA was also found to enhance cell adhesion and osteogenic differentiation in human mesenchymal stem cells [29,30]. TW1-NIRF is applied to monitor osteoblastic activity. This monitoring imaging system can trace/track cells differentiated within bone in vitro and in vivo in real time and can also quantify osseointegration. This novel method may provide a multimodality imaging where other imaging modalities can be employed simultaneously. Thus, this new imaging method can repair the artifacts caused by computed tomography.

## 2. Results

### 2.1. Animal Model Design and Imaging Evaluation of Implants

A titanium alloy screw was implanted in the anterior hard palate of a 6-month-old Sprague Dawley rat (Figure 1B). The sagittal images were taken from CT and X-ray of the titanium screw–implanted (Figure 2A) and sham-operation (Figure 2B) rats 28 days after surgery. The X-ray showed that the tissue surrounded the implant tightly. No gap was found between the bone tissue and the implant. However, a gap of metal artifacts was noted around the interface between the dental implants and rat palatal bone 28 days after the surgical implantation in micro-CT images (Figure 2C,D).

### 2.2. In Vivo Imaging

The TW1-NIRF image system specifically revealed the implantation site to present a significant NIRF signal. Metal artifacts appeared around the interface between the dental implants and rat palatal bone in micro-CT images (Figure 2E). The NIRF and merged TW1-NIRF images demonstrated that the probe specifically accumulated at the implantation site (Figure 2F). The areas of high HA synthesized around the implant are visible. The signal intensity exhibited in the sham operation group compared to the titanium screw-implanted group (1.00 ± 0.028 to 3.02 ± 0.546, *p* < 0.005) is shown in Figure 2G. The 3-fold augmented fluorescence intensity indicates the probe’s targeting of new bone formation sites around the implant.

### 2.3. Histological and NIRF Analysis of Perio-Implant Tissue

To further confirm the ability of pamidronate-NIRF probes to detect new bone formation in vivo, rats were sacrificed for NIRF imaging and the histological analysis of the maxilla bone. The dissected maxilla bone was sectioned in different planes to evaluate the probe deposition at the new bone formation site around the implantation site (Figure 3A). The longitudinal (Figure 3B,C) and horizontal section (Figure 3D,E) images of the maxilla tissue also showed positive-probe-stained tissue around the implant, indicating that the pamidronate-NIRF probe was able to detect bone tissue formation around the implantation site.

## 3. Discussion

Colloid gold has been widely used in biomedical detection due to its strong surface plasmon absorption. In the last 10 years, gold particles have been used as a contrast agent to improve X-ray or MRI imaging [31,32]. Due to its biocompatibility and stable optical properties, colloid gold has become an alternative to quantum dots for in vivo application [33]. Gold nanoclusters have several advantages for molecular image application: a decent quantum yield, a highly colloidal stability, and an extremely small size (core diameter <1.5 nm); furthermore, they can also be conjugated with biological molecules. Moreover, the quantum yield of gold nanoclusters can be further improved through thermal treatment and biomolecule encapsulation. As a result, we hypothesized that pamidronate conjugated gold nanoclusters can be utilized to monitor the new bone formation. Therefore, the new bone, surrounding the implant, can be detected.

Our previous data showed that near-infrared (NIR) fluorescent probes can be used to monitor the in vitro differentiation of human mesenchymal stem cells into osteoblasts as well as the osteogenesis process at a cellular level [12]. In this study, we designed a new imaging system for the in vivo detection of NIR signals in rats, which are much larger than mice in size. Most studies examine NIR signal intensity using mice or nude mice, not only because of the easy manipulation of NIR signal penetrating into the subepithelial layer, but also because fluorescence signals exhibits are visible from the mouse body. 

In this study, we used rats as the model, which have 10–12 times greater body weight compared to the mice. To successfully utilize this model, the newly designed system contains a back-illuminated Peltier 16-bit charge-coupled device (CCD, −90 °C cooled) (Hammamastu, Hammamastu City, Japan) with a pixel size of 24 μm × 24 μm, 230,000 electrons, and 8 × 8 binning. This imaging achieves >80% of the quantum efficiency at an emission of 800 nm. Thus, this allows the camera to identify the osseointegration of the implant within the bone tissue.

The lesion site in the rat reveals a significant fluorescence after the injection of the 20 nM nano-Au-Pam probe. Previous studies in other laboratories showed that an Au NP probe dose of 1 mg/Kg was administered in rats [34]. Our previous data also demonstrated a dosage of 2 nM for nude mice. The weight of the rat is 10–12 times than the nude mice. The Au NPs are considered to preserve a lower toxicity compared to the Cat B probe containing Cy5.5 [35]. Furthermore, an MTT experiment showed that the Au NPs exhibit a better viability compared to QD [36]. These indicate that a minimal dose can induce a significant signal intensity in this animal model [34,35,36]. Despite the fact that 20 nM is a minimal dose, the dosage should be reduced and the signal intensity should be increased before the NIR probes are applied in the clinic.

For the clinical translation, the NIR probe requires a decrease in the dose, an increase in the optical stability, selective targetability, reliable pharmacokinetic activity, complete clearance, and nontoxicity. Incomplete clearance will cause potential toxicity through biological interactions in the body. In order to overcome the clinical limitations, an NIR probe is necessary to consider the target specificity and biodistribution. 

In this study, NIRF signals indicating new bone formation were observed around the implantation site in a maxilla implantation model, demonstrating a significant difference between the new bone formation site around the metal implant and normal tissue. This novel nano-Au imaging system can be applied to monitoring bone maturation around orthopedic implants or quantifying the osseointegration of bone and dental implants. Therefore, by observing the process and activity of osteoblasts, we can establish a new implant loading timetable. New types of implant fixtures or surface treatments can also be evaluated through this system. In addition, according to this characteristic of observing the bone formation process, the system may be further applied in the early diagnosis of prostate cancer and other cancer metastases.

## 4. Materials and Methods

### 4.1. Probe Synthesis

A gold nanocluster-based pamidronate NIR fluorescent probe was synthesized according to the methods in our other studies [37,38,39]. Briefly, fluorescent gold nanoclusters were synthesized using nanoparticle-etching methods. Pamidronate was attached to the carboxylated surface of a nanocluster using an N-(3-dimethylaminopropyl)-N′-ethylcarbodiimide (EDC) (Sigma-Aldrich, St. Louis, MO, USA) crosslinker. Briefly, nano-Au-Pam (em 674 nm) was synthesized by immobilizing pamidronate (Sigma 2371) onto nano-Au particles using a zero-length crosslinking agent, 1-ethyl-3-[3-dimethylaminopropyl] carbodiimide hydrochloride (EDC). To link the pamidronate to nano-Au particles, equal volumes of Au nanoclusters (40 mM), pamidronate (3 mM), and EDC (80 mM) were mixed at room temperature for 2 h for the crosslinking reaction. To concentrate the conjugated nano-Au-Pam, the reaction solution was loaded in a 30-kDa molecular sieve (Amicon, Millipore, Bedford, MA, USA) and centrifuged to concentrate the conjugated nano-Au-Pam at 3000 rpm for 10 min. The solution was centrifuged twice and washed with phosphate-buffered saline (PBS) (Sigma-Aldrich, MO, USA) to remove any unbound reactants.

### 4.2. Animals

A bone defect model for the NIRF probe detection in rats was established. Briefly, 12 6-month-old Sprague Dawley rats were housed in a laboratory at 22.2 °C under a 12-h light and 12-h dark cycle and fed ad libitum. The animal study was approved by the Taipei Medical University Institutional Animal Care and Use Committee (LAC-2014-0391) and complied with the committee’s regulations. All animal husbandry and handling procedures including animal monitoring, diet, primary enclosures, and environmental control followed standard operating procedures in accordance with the Animal (Scientific Procedures) Act of 1986.

### 4.3. Experimental Design and Surgical Techniques

The 12 Sprague Dawley rats included 3 groups: 4 rats in the X-ray and the (computed tomography) CT group, 4 rats in the NIRF group, and 4 rats in the sham group. Animals were anesthetized with Chanazine 2% dilute with 10× PBS 10 μL + Zoletil 50 dilute 2× PBS 10 μL with or without implantation by titanium alloy screws (1.2 mm [diameter] × 2.4 mm [length], Ti-6AL-4V ELI, Self-Drilling Bone Screw System, ACE Surgical Supply Co., Inc., Brockton, MA, USA) in the anterior hard palate. Rats were monitored using X-ray, microcomputed tomography, and NIRF imaging 28 days after the surgery (Figure 1A,B). After imaging evaluation, the rats were sacrificed and the maxilla was dissected and fixed in 10% buffered formalin for NIRF imaging and histological analysis.

### 4.4. In Vivo Imaging

Animals were examined using microcomputed tomography (micro computed tomography [CT]; Triumph XO CT System, Chatsworth, CA, USA) and optical imaging (TW1-NIRF; Taipei, Taiwan) 28 days after surgery. For micro-CT scans, rats were maintained under general anesthesia during the scanning procedure. Each rat was placed in a sample holder in the cranial–caudal direction and scanned using a high-resolution micro-CT system at a spatial resolution of 80 μm (voxel dimension) and in 1024 × 1024-pixel matrices. The animals were evaluated using both micro-CT and X-ray to confirm the success of the implantation. The bone tissue was segmented using a global thresholding procedure. The threshold was set to 1600 unit to investigate the peri-implant bone tissue.

The TW1-NIRF imaging system (Figure 1C), designed and assembled in our laboratory, contains a back-illuminated Peltier 16-bit charge-coupled device (CCD, −90 °C cooled) (Hammamastu, Hammamastu City, Japan) with a pixel size 24 μm × 24 μm. This imaging gains >80% of quantum efficiency at emission 800 nm. The animals were injected with 20 nM of nano-Au-Pam probe systemically. Two hours after injection, the animals were anesthetized with isofluorane and subjected to NIRF imaging [40]. The TW1-NIRF system contained a 150 W halogen lamp, and an excitation, and an emission bandpass filter of 800 nm and 610–650 nm, respectively. Whole body NIRF images were obtained with acquisition times of 2 min, and the white light images were obtained in 0.075 s. HCImage Live software was used to compile the images (Hamamatsu, Sewickley, PA, USA).

In order to determine whether the osseointegration occurred in the metal artifacts, the NIRF image was compared to micro-CT in the same metal artifact areas. To visualize the fluorescent signal emitted from the probe in vivo, the animals were injected with 20 nM nano-Au-Pam probe and evaluated using the TW1-NIRF system for NIRF imaging, then followed by micro-CT. The ICG channel (excitation wavelength of 710–760 nm and emission wavelength of 810–875 nm) was used in the TW1-NIRF system for detection.

### 4.5. Histological and NIRF Analysis

After imaging evaluation, animals were sacrificed and the hard palate tissue was dissected for histological analysis. The hard palate tissue was fixed and embedded in paraffin. After the microtome section, HE staining and mounting were used. Longitudinal and horizontal section tissue samples were observed by NIRF (IR800 filter) to detect the visible areas of high HA synthesized around the implant.

### 4.6. Statistical Analysis

To quantitate the NIRF signal, a circular region-of-interest (ROI) was manually defined around the implant area, and the average signal within the ROI was obtained. Data are presented as mean ± SD. A two-tailed Student’s *t*-test was conducted. *p* < 0.05 was considered significant.

## 5. Conclusions

Taken together, the results showed the potential application of this novel nanogold-based NIRF probe system for assessing changes in bone tissue formation after implantation. Using the probe design and image system as a prototype, this image platform could be further developed to repair the artifacts caused by computed tomography and upgrade the evaluation of dental implantation failure by detecting bone formation in situ.

## Figures and Tables

**Figure 1 ijms-24-05032-f001:**
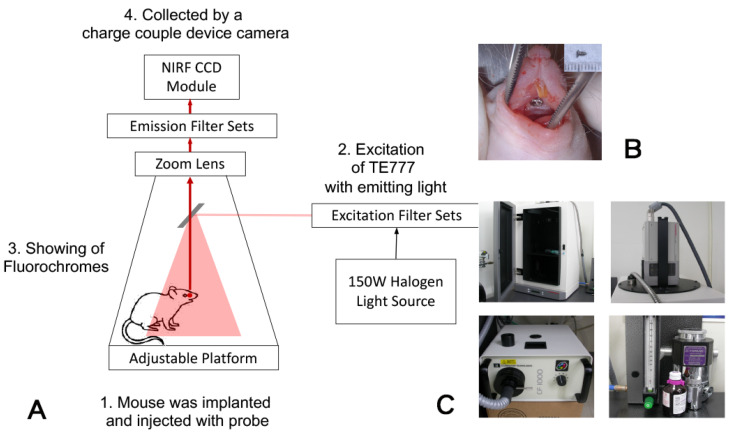
Illustration of monitoring osteogenesis and osseointegration using near-infrared fluorescent imaging. (**A**) The optical design of TW1-NIRF system. Briefly, the excitation wavelength of 800 nm was generated from a halogen light source. Study animals were gas-anesthetized with isoflurane and placed in the imaging chamber. The emission was 610–650 nm and was captured by a high-sensitivity CCD. (**B**) A titanium alloy screw was implanted in the anterior hard palate of 6-month-old Sprague Dawley rats. Titanium alloy screws (1.2 mm [diameter] × 2.4 mm [length], Ti-6AL-4V ELI, Self-Drilling Bone Screw System, ACE Surgical Supply Co., Inc., Brockton, MA, USA). (**C**) The TW1-NIRF imaging is composed of an imaging chamber, NIRF CCD module with lenses and a filter wheel, excitation light source, and gas anesthesia module.

**Figure 2 ijms-24-05032-f002:**
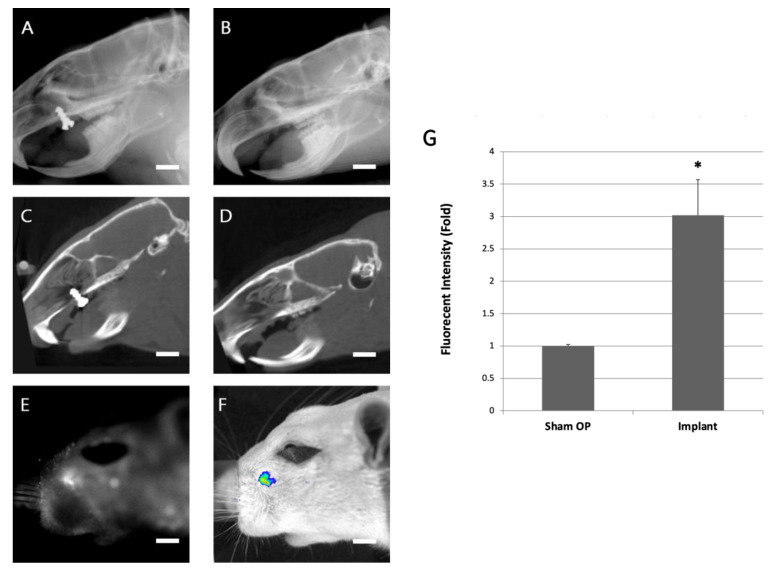
Animal model for computed tomography metal artifact adjustment. The sagittal X-ray of the (**A**) titanium screw–implanted and (**B**) sham-operation rats 28 days after surgical implantation. The micro-CT images of the (**C**) titanium screw implanted and (**D**) the sham-operation rat 28 days after surgical implantation. In the sagittal X-ray image, the bone tissue surrounded the implant tightly and no gap was found between the bone tissue and the implant. However, in the micro-CT image, a gap of metal artifacts was noted around the interface between the dental implant and the rat palatal bone. (**E**) The fluorescence image was obtained using pamidronate-NIRF probes in the titanium screw-implanted group. (**F**) A merged in vivo image was used to improve the visual result in the titanium screw-implanted group. The scale bar represents 2.5 mm in (**A**–**F**). (**G**) The fluorescent intensity was quantified in the sham operation group versus the titanium screw-implanted group (right column chart). The error bars represent the standard deviation at each data point. The asterisk (*) indicates the statistical significance of differences at *p* < 0.005.

**Figure 3 ijms-24-05032-f003:**
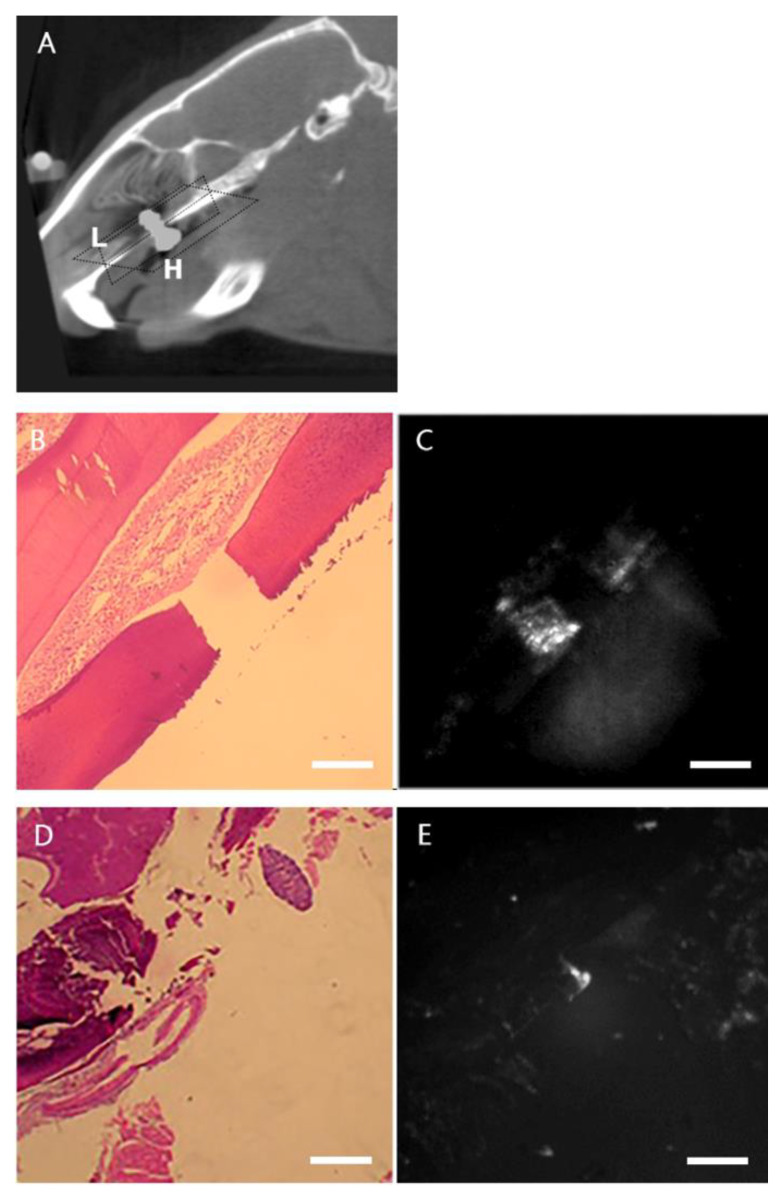
Evaluation of the correlation between histology and NIR fluorescence of peri-implant tissue. (**A**) Longitudinal and horizontal sections of pamidronate-NIRF probe injected 28 days after implantation. L = longitudinal section and H = horizontal section. (**B**) The peri-implant tissue was longitudinally excised en bloc. A gap of implantation is shown in the HE staining at ×4 magnification. (**C**) Fluorescence signal is noted at the peri-implant area (×4 magnification). (**D**) Horizontal section of the en bloc of the peri-implant tissue. An implant defect is shown in the HE staining at ×2 magnification. (**E**) The fluorescence signal is seen at the peri-implant area (×2 magnification). The scale bar represents 1 mm in (**B**,**C**) and 2 mm in (**D**,**E**). The fluorescence indicates the areas of high HA synthesized around the implant.

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
