# Peer review of "A Non-Invasive Method for Monitoring Osteogenesis and Osseointegration Using Near-Infrared Fluorescent Imaging: A Model of Maxilla Implantation in Rats"

_ijms, 2023, doi:10.3390/ijms24055032_

Round 1

Reviewer 1 Report

This study aims to investigate an interesting topic which is pertinent to the oral health research, the experiment as well as the reporting and interpretation of data are well-executed.

Please find my comments as follows:

-The entire manuscript needs a grammatical revision. For instance: P.2 L.53 “..NIRF imaging technologies was (were) developed…”; L 59…  binds (to)… ; L124.. maxilla (was) dissected…, and etc.

Introduction:

-Can you expand what “TW1-NIRF” stands for?

Methods:

-Ethical approval statement is missing.

- Information on how the fluorescent intensity was quantified is not given.

-No information on statistical analysis done for fluorescent intensity is provided.

Results:

-A,B,C, D are missing from Figure 2-3.

-Lines 162-164 should be in the methods section. “The animals were evaluated by using micro-CT and X-162 ray to confirm the implantation’s success. The bone tissue was segmented using a global 163 thresholding procedure. The threshold was set to 1600 unit to investigate the peri-implant 164 bone tissue.”

-Same for Lines 184-189. They should be presented in the methods.

-Explanation of graph 2G should be presented in the text and not just in the figure caption.

Discussion:

-Discussion needs to be expanded. Please discuss the applicability of translating this novel nanogold-based NIRF probe system into human structures. It is worthy to mention the non-invasiveness of this method as well.

-Limitations should be added in the discussion part.( e.g. considering the 24h wait time)

Author Response

Dear reviewer:

Many thanks for the comments. I revised and English edited all manuscript as the followings:

Reviewer 1.
This study aims to investigate an interesting topic which is pertinent to the oral health research, the experiment as well as the reporting and interpretation of data are well-executed. 

Please find my comments as follows:

-The entire manuscript needs a grammatical revision. For instance: P.2 L.53 “..NIRF imaging technologies was (were) developed…”; L 59…  binds (to)… ; L124.. maxilla (was) dissected…, and etc.

Ans. Revised all following the instruction.

Introduction:

-Can you expand what “TW1-NIRF” stands for?

Ans. Taiwan 1 (TW1)-NIRF. Thanks!

Methods:

-Ethical approval statement is missing.

Ans. Added as the instruction. ….The animal study was approved by the Taipei Medical University institutional Animal Care and Use Committee (LAC-2014-0391) and complied with the committee’s regulations.

- Information on how the fluorescent intensity was quantified is not given.

Ans. Added as the instruction.

-No information on statistical analysis done for fluorescent intensity is provided.

Ans. Added the statistical analysis section. Many thanks!

Results:

-A,B,C, D are missing from Figure 2-3. Ans. Added as the instruction.

-Lines 162-164 should be in the methods section. “The animals were evaluated by using micro-CT and X-162 ray to confirm the implantation’s success. The bone tissue was segmented using a global 163 thresholding procedure. The threshold was set to 1600 unit to investigate the peri-implant 164 bone tissue.” Ans. Revised as the instruction.

Same for Lines 184-189. They should be presented in the methods.

- Ans. Revised as the instruction.

-Explanation of graph 2G should be presented in the text and not just in the figure caption.

- Ans. Revised as the instruction.

Discussion:

-Discussion needs to be expanded. Please discuss the applicability of translating this novel nanogold-based NIRF probe system into human structures. It is worthy to mention the non-invasiveness of this method as well.

Thanks to comments. Revised as the following

The lesion site reveals a significant fluorescence after the injection of the 20 nM nano-Au-Pam probe. This indicates that a minimal dose can induce a significant signal intensity in the animal model [38-40]. The NIR probe dosage can be further reduced to be used in clinic.

For the clinical translation, the NIR probe requires a decrease in the dose, an increase in the optical stability, selective targetability, reliable pharmacokinetic activity, complete clearance, and nontoxicity. Incomplete clearance will cause potential toxicity through biological interactions in the body. In order to overcome the clinical limitations, an NIR probe is necessary to consider the target specificity, and biodistribution.

-Limitations should be added in the discussion part.( e.g. considering the 24h wait time)

Ans. Some errors. Revised and referenced. “Two hours after injection, the animals were anesthetized with isofluorane and subjected to NIRF imaging [34].”

Reviewer 2 Report

Authors mention in the Introduction the importance of a better approach to monitor osteogenesis and osteointegration due to many reasons such as the use of implants during the past ten years or so, but later in the manuscript they don’t refer to this anymore as an interpretation or link between their study and the current needs.  

In the Materials and Methods section, authors describe that animals were injected with 20 nM of nanoAu Pam probe, are there any risks or potential toxicity? How can this be monitored? 

In the Results section there are some issues with figures. For example, Figure 2 doesn’t have the labels for A-D. Also, figure G is hard to see as it is too small. Do error bars represent standard deviation? If yes, how many samples were used to calculate error bars? Also, images don’t have scale bar at all. 

Figure 3 doesn’t have the labels for A-E. None of them include scale bars.

In the Discussion section, lines 215-239 are more likely to be in the Introduction than in this section.

In lines 240-241 authors say that the lesion site reveals a significant fluorescence after injection of 20 nM nanoAu Pam 240probe. This indicates that a minimal dose can induce a significant signal intensity in clinic. How can authors calculate the optical dose required? Is there a range for this? What are the criteria?

In References, most of them are not from recent years, so increasing the number of references to include more recent ones would be beneficial.

Overall: The study includes a novel approach, but the different sections require modifications/changes in order to improve the manuscript. Specifically, the Discussion section needs to be improved because there is no clear explanation and interpretation of results compared to similar studies, for example using UV or vis light ranges. Additionally, an interpretation of how this technique could be used in humans needs to be incorporated, highlighting the advantages and limitations. Finally, it is important to compare results with a control to determine the main differences. 

Author Response

Dear Reviewer:

Thank you for spending time with me to correct our manuscript. We revised and English edited all manuscript as followings:

Reviewer 2.

Authors mention in the Introduction the importance of a better approach to monitor osteogenesis and osteointegration due to many reasons such as the use of implants during the past ten years or so, but later in the manuscript they don’t refer to this anymore as an interpretation or link between their study and the current needs.  

Ans. Thank you for the comment.

We added in the introduction section “The novel method may provide a multimodality imaging where other imaging modalities can be employed simultaneously. Thus, the new imaging can repair the artifacts caused by computed tomography.”

Also in the conclusion“…………to repair the artifacts caused by computed tomography and upgrade the evaluation of dental implantation failure by detecting bone formation in situ.”

In the Materials and Methods section, authors describe that animals were injected with 20 nM of nanoAu Pam probe, are there any risks or potential toxicity? How can this be monitored? 

Ans. Thanks to the comment.

Ref. 38 taught us probe dose is at1 mg/Kg. Ref. 39 taught us 2nM for nude mice. The rat is 10-12 time than the nude mice. Furthermore, the nano-Au probe preserves lower toxicity compared to the Cat B probe containing Cy5.5 (Ref. 39).

Ref. 40 demonstrated a MTT experiment. This taught us nano-Au exhibits a better viability compared to QD.

In the Results section there are some issues with figures. For example, Figure 2 doesn’t have the labels for A-D. Also, figure G is hard to see as it is too small. Do error bars represent standard deviation? If yes, how many samples were used to calculate error bars? Also, images don’t have scale bar at all. 

 Ans.  Followed the instruction and added the bars.

Figure 3 doesn’t have the labels for A-E. None of them include scale bars.

Ans. Label followed the instruction.

In the Discussion section, lines 215-239 are more likely to be in the Introduction than in this section.

Ans. Many thanks for the comments.  We have rewritten and reorganized the discussion section.

In lines 240-241 authors say that the lesion site reveals a significant fluorescence after injection of 20 nM nanoAu Pam probe. This indicates that a minimal dose can induce a significant signal intensity in clinic. How can authors calculate the optical dose required? Is there a range for this? What are the criteria?

Ans. Thank you for the crucial comment.

The molecular weight of the nano-Au-Pam probe is 10-30 Kda.

10-30 Kda means 10000-30000 g/mole.

20 nM contains 10000-30000 g/mole x (20x 10-9 mole) = 2-6 x 10-4 g. The dose is small.

Ref. 38 taught us probe dose is at1 mg/Kg. Ref. 39 taught us 2nM for nude mice. The rat is 10-12 time than the nude mice. Furthermore, the nano-Au probe preserves lower toxicity compared to the Cat B probe containing Cy5.5 (Ref. 39).

Ref. 40 demonstrated a MTT experiment. This taught us nano-Au exhibits a better viability compared to QD.

In References, most of them are not from recent years, so increasing the number of references to include more recent ones would be beneficial.

Ans. Thank you for the suggestion. Added some references in recent years. Seven references in 3 years, and more than half references after 2010.

Overall: The study includes a novel approach, but the different sections require modifications/changes in order to improve the manuscript. Specifically, the Discussion section needs to be improved because there is no clear explanation and interpretation of results compared to similar studies, for example using UV or vis light ranges. Additionally, an interpretation of how this technique could be used in humans needs to be incorporated, highlighting the advantages and limitations. Finally, it is important to compare results with a control to determine the main differences. 

Ans. Revised as instruction. Added or revised or English editing through all manuscript.

Reviewer 3 Report

I agree with the authors that near-infrared ray fluorescent (NIRF) imaging does have the ability to be a useful tool with the capacity to overcome current diagnostic limitations. The results, if presented in a more suitable manner, would be suitable for publication and do show the benefits of NIRF. However, in its current form the paper is not in a position to be published.

The discussion section contains little to no discussion regarding the results provided. NIRF images are poorly described and the paper lacks informative discussion of a comparison with additional techniques applied. Overall the paper is composed in the manner of a laboratory report and does not feature the insightful analysis and discussion that would be expected in an academic paper. The figures are difficult to interpret and are not labelled correctly. This makes reviewing the work difficult. The conclusions are too brief and do not accurately highlight the contents and importance of the study. As the paper contains many grammatical and tense related errors with abbreviations often incorrectly assigned, it would require extensive proof editing prior to any future submission.

Minor

Abstract need to be condensed. Currently too long and repetitive.

Line 32: State "hydroxyapatite"

Line 53: State "near-infrared ray fluorescent" for first time use within the manuscript (not including abstract).

Line 58: What is meant by "prime" material product?

Line 66: Reference required.

Line 76: Please be consistent with the use of italics.

Line 84: What does TW1 stand for?

Figure 1: Poor image formatting. Images overlap and text is blurred. Please increase image resolution.

Figure 2: Poor image formatting. Unable to see A – D labels. The graph is too small to read axis labels. How do you explain the large SD in the Sham and Implant group?

Figure 3: Cannot follow figure with suitable figure labels. No scale bars.

Author Response

Dear Reviewer:

Much appreciation for your comments and mentor. We revised and English edited all manuscript as the followings:

Reviewer 3.

I agree with the authors that near-infrared ray fluorescent (NIRF) imaging does have the ability to be a useful tool with the capacity to overcome current diagnostic limitations. The results, if presented in a more suitable manner, would be suitable for publication and do show the benefits of NIRF. However, in its current form the paper is not in a position to be published.
Ans. Thanks for the comments. Revised as the instruction.
The discussion section contains little to no discussion regarding the results provided. NIRF images are poorly described and the paper lacks informative discussion of a comparison with additional techniques applied. Overall the paper is composed in the manner of a laboratory report and does not feature the insightful analysis and discussion that would be expected in an academic paper. The figures are difficult to interpret and are not labelled correctly. This makes reviewing the work difficult. The conclusions are too brief and do not accurately highlight the contents and importance of the study. As the paper contains many grammatical and tense related errors with abbreviations often incorrectly assigned, it would require extensive proof editing prior to any future submission.
Ans. Added or revised or English editing following the comments through all manuscript.
Minor

Abstract need to be condensed. Currently too long and repetitive.
Ans. Revised as the instruction. Many thanks!
Line 32: State "hydroxyapatite"
Ans. State “hydroxyapatite” in the introduction section, and removed the HA from the abstract section.
Line 53: State "near-infrared ray fluorescent" for first time use within the manuscript (not including abstract).

Ans. State as the instruction. Thanks!

Line 58: What is meant by "prime" material product?
Ans. I meant “Major”. May be “prime” is not clear, already remove. Thanks!

Line 66: Reference required.
Ans. Added ref. 6 and 7. Thanks!
Line 76: Please be consistent with the use of italics.
Ans. Revised as the instruction.
Line 84: What does TW1 stand for?
The NIRF system was designed and assembled in our lab and we named the system TW-1, which means Taiwan-1.

Figure 1: Poor image formatting. Images overlap and text is blurred. Please increase image resolution.
Figure 1A-C is updated with high resolution TIFF.

Figure 2: Poor image formatting. Unable to see A – D labels. The graph is too small to read axis labels. How do you explain the large SD in the Sham and Implant group?
Figure 2A-G is updated with high resolution TIFF and labeled with A-G. The axis labels are revised accordingly. The large standard deviation in the implant group is due to that the fluorescent intensities are measured from 3 individual in vivo experiment repeats and the fluorescent intensity readings are easily affected by the sample placement (experiment animal under anesthesia) in the imaging chamber. In contrast, the standard deviation in the sham OP group is small because the data only reflect the background autofluorescence. Even though, the average fluorescent intensity of implant group shows still 10-fold increase as compared to the Sham OP group, indicating that the probe does accumulated at the implantation site and can be effectively detected by the NIRF system.

Figure 3: Cannot follow figure with suitable figure labels. No scale bars.

Figure 3 is labeled with A-E and the scale bars are added. The scale bar represents 1-mm in Figure 3B and 3C (x4 magnification), and 2-mm in 3D and 3E (x2 magnification).

Round 2

Reviewer 2 Report

In figure 4, why do you have such variation? It is almost the same as the entire bar, is 3 repeats enough? 

In the Discussions section, authors need to include a more detailed explanation of the potential use of this approach in humans. Also references supporting their assumptions. For example,  Why a decrease in the dose is needed? What is the theoretical dose range and why? 

Author Response

Minor:

Figure 2 and 3: Please add scale bars. Current scalebars in Figure 3 are too small and I cannot the units on the bar.

Many thanks for the comments. The scale bars are added in Fig. 2.

We also increase the resolution of Fig. 3 from 300 to 400 pixels, and enlarge the size to 12.4 cm x 8 cm.

Figure 3: Consider reformatting figure to enlarge images.

We re-formate and enlarge images of Fig. 3 as the instruction.

Reviewer 3 Report

Minor:

Figure 2 and 3: Please add scale bars. Current scalebars in Figure 3 are too small and I cannot the units on the bar.

Figure 3: Consider reformatting figure to enlarge images.

Author Response

In figure 4, why do you have such variation? It is almost the same as the entire bar, is 3 repeats enough? 

Much appreciation for the comments. Some errors were corrected. The animal number is 4 in each group (n=4). Three-repeat was performed. Therefore, the new results were shown as (1.00+0.028 to 3.02+0.546, P<0.005).  

In the Discussions section, authors need to include a more detailed explanation of the potential use of this approach in humans. Also references supporting their assumptions. For example, Why a decrease in the dose is needed? What is the theoretical dose range and why? 

Thanks to the constructive comments. We explain the potential clinical use, and the theoretical dose range as the instructions.

"The lesion site in the rat reveals a significant fluorescence after the injection of the 20 nM nano-Au-Pamprobe. Previous study in other laboratory showed that an Au NP-probe dose of 1 mg/Kg was administered at rats [38]. Our previous data also demonstrated a dosage of 2nM for nude mice. The rat is 10-12 time than the nude mice. The Au NPs is considered to preserve a lower toxicity compared to the Cat B probe containing Cy5.5 [39]. Furthermore, a MTT experiment showed that the Au NPs exhibit a better viability compared to QD [40]. These indicate that a minimal dose can induce a significant signal intensity in this animal model [38-40]. Despite of the fact that 20 nM is a minimal dose, the dosage should be reduced and the signal intensity should be increased before the NIR probes applied in clinic."